# Effects of a time-use intervention in isolated patients with coronavirus disease 2019: A randomized controlled study

Jae Hyu Jung[ID][1], Jin Young Ko[2], Ickpyo Hong[3], Min-Ye Jung[3], Ji-Hyuk Park[ID][3]*

1 Department of Occupational Therapy, Gyeonggi Provincial Medical Center, Gyeonggi-do, Republic of Korea, 2 Department of Public Health Care (Rehabilitation), Seoul National University Bundang Hospital, Seongnam-si, Republic of Korea, 3 Department of Occupational Therapy, Yonsei University, Wonju, Republic of Korea

* otscientist@yonsei.ac.kr

## Abstract

### Objective

Patients with coronavirus disease experience deterioration in occupational balance and mental health. The primary objective of this study was to determine the effectiveness of a time-use intervention on the occupational balance of isolated patients with coronavirus disease. Its impact on secondary outcomes including mental health and quality of life was also assessed.

### Methods

This randomized controlled clinical trial was conducted in a single community-based hospital. Forty-one patients (19 in the experimental group and 22 in the control group) with coronavirus disease were recruited between February 1, 2021, and March 19, 2021. Participants were randomly assigned to receive a time-use intervention or education on self-activity. The time-use intervention is to plan a daily routine to engage in meaningful occupations. It consisted of 4 steps: time-use analysis, occupation selection, arrangement of activities and practice, and occupational therapist intervention. The control group was educated on self-activity and spent time autonomously.

### Outcomes and measures

The primary outcome was occupational balance, evaluated using the Korean version of the Life Balance Inventory. Secondary outcomes were mental health and quality of life assessed using the Korean version of the Patient Health Questionnaire-9, Korean Form of Zung's Self-Rating Anxiety Scale, Korean version of the Insomnia Severity Index, Multidimensional State Boredom Scale-8, Fear of Coronavirus Disease: Korean version of the Fear of Coronavirus Disease Scale, and World Health Organization Quality of Life Assessment Instrument-BRIEF. Outcome measures were evaluated at admission and discharge.

**Data Availability Statement:** All relevant data are within the paper and its Supporting Information files.

**Funding:** This research was supported by the Ministry of Education of the Republic of Korea and the National Research Foundation of Korea (NRF-2021S1A3A2A02096338)

**Competing interests:** The authors have declared that no competing interests exist.

## Results

The time-use intervention significantly improved occupational balance ($F$ = 14.12, $p$ < .001) and all other measures of depression, anxiety, boredom, fear, and quality of life. Conversely, the control group showed a worsening pattern for all measures.

## Conclusion

The time-use intervention is effective for improving occupational balance, mental health, and quality of life in patients with coronavirus disease.

## Introduction

The World Health Organization declared the coronavirus disease (COVID-19) pandemic on March 11, 2020, and the new coronavirus has spread worldwide. To prevent transmission, countries are screening for and isolating patients with COVID-19. In Korea, patients are isolated in hospitals or community treatment centers [1] for at least 10 days based on isolation release criteria [2]. Isolation greatly helps reduce viral transmission [3]. However, it can change an individual's daily life and cause major disruptions [4].

Occupation balance is defined as the organization of daily activities that facilitate health and well-being by allowing patients to participate in various activities such as work, labor, household management, childcare, leisure, and rest. Occupation imbalance causes negative psychological states such as significant stress and mental health problems [5]. Occupation changes have also been observed in patients with COVID-19, and they persist in some patients even after discharge [6,7].

Time use is closely related to occupational balance because it is fundamental to how people organize and structure their daily lives [8]. Understanding how people spend their time can be a particularly useful approach in work balance and engagement studies [9]. A time-use intervention minimizes meaningless time through time planning and enables the efficient use of time. Time-use interventions aim to maintain health and well-being by properly distributing time within the occupation area to maintain occupation balance [8,9]. Time-use interventions have been applied to patients with a variety of diagnoses and have shown positive effects on occupational balance, mental health, and quality of life (QOL) [8,10,11].

Occupational imbalance and mental health problems have been reported in patients with COVID-19; however, studies on occupational balance are lacking. Therefore, this study aimed to investigate the effect of a time-use intervention on occupational balance, mental health, and QOL in patients with COVID-19 who were forced to isolate.

## Materials and methods

### Ethics statements

The study was registered with cris.nih.go.kr prior to initiation (identification number: KCT0005711) and approved by the local institutional review board of Yonsei University Mirae Campus (1041849-202102-BM-024-03). The requirement for written informed consent was waived owing to the risk of virus transmission. As an alternative to written consent, online consent was obtained to proceed with the pre-evaluation; patients clicked the "I agree" button on the Google form using a tablet PC.

## Study design and patients

Fifty patients with confirmed COVID-19 admitted to a single community-based hospital were recruited for this randomized controlled clinical trial study. Participants were randomized into experimental and control groups, with 25 participants in each group, using a block randomization method (Fig 1). The two groups were assigned to wards separately so that they did not mix. Block sizes were 6, 8, 10, 12, and 14. This study was a non-blinded trial, and the participants were assigned in the order of beds by a physician after checking their medical state. The ward nurse routinely conducted medical checkups, such as chest radiography, blood pressure, blood tests, and oxygen saturation for all inpatients, regardless of the research study. This was done to determine the severity of pneumonia and whether the patient was hemodynamically stable. Only patients who agreed to participate in the study had their medical status reviewed by the medical staff of this study. The inclusion and exclusion criteria were as follows.

## Inclusion criteria

- Clinically diagnosed with COVID-19

- No underlying disease of the respiratory system

- Oxygen saturation of ≥95% and hemodynamically stable

- At least 18 years of age

- Patient understood the purpose of the research and provided consent

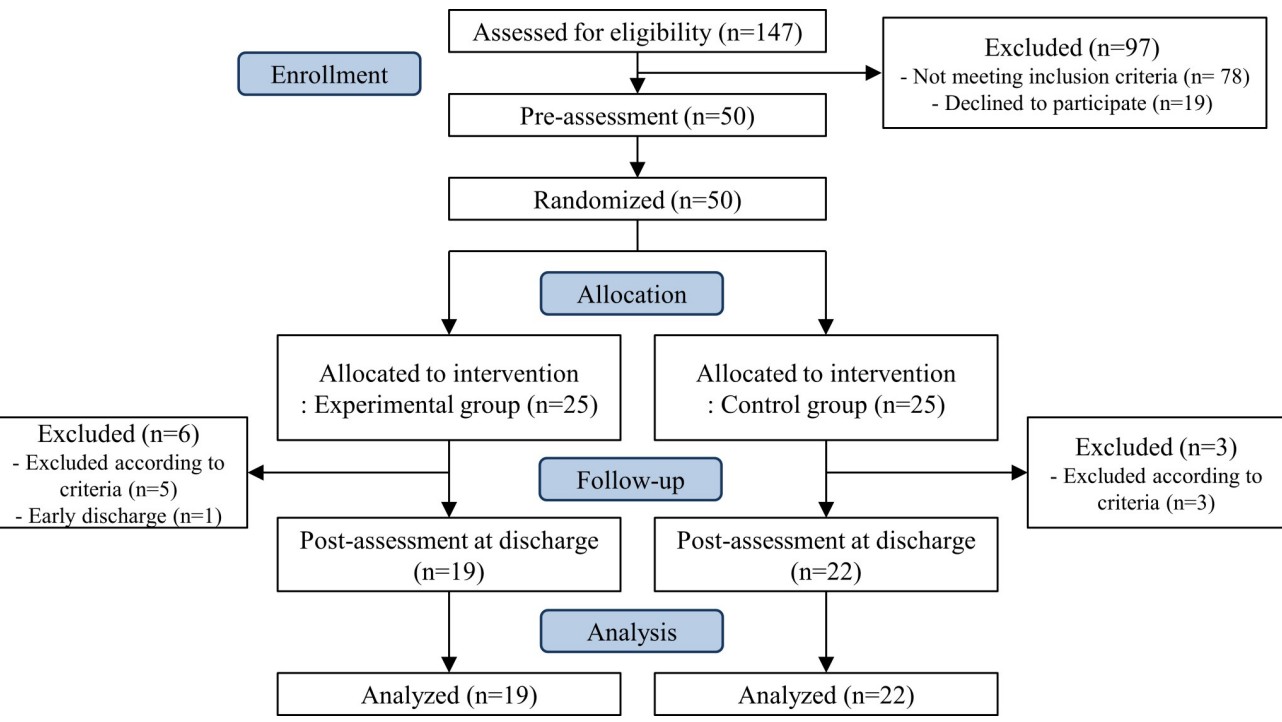

**Fig 1. CONSORT flow diagram of recruitment, allocation, and participation.**

### Exclusion criteria

- Patients who were medically unstable

- Patients who had difficulties in reading and understanding the Korean language or had communication problems

### Intervention protocol

The experimental group underwent a time-use intervention, whereas the control group received education for self-activity. The interventions were performed daily during the isolation period. All participants received the intervention individually.

### Time-use intervention

The time-use intervention was performed over 7 days, depending on the length of hospitalization in the experimental group. The protocol used in this study (Fig 2) consists of 4 steps. The first step is the time-use analysis. Patients and the therapist analyzed the activities of the previous day and the planned time schedule. The next step was the occupation selection. Occupation selection was based on the Korean version of the Life Balance Inventory (K-LBI) results. There were 53 occupations in the K-LBI group. Activities in which the patient was doing less than the desired time and activities in which the patient was interested were added. Through interviews, we selected possible occupations in the isolation ward. The third step was the arrangement of activities. Patients and the therapist placed meaningful tasks in the meaningless time selected in step 2. In addition, patients and the therapist created a timetable. Steps

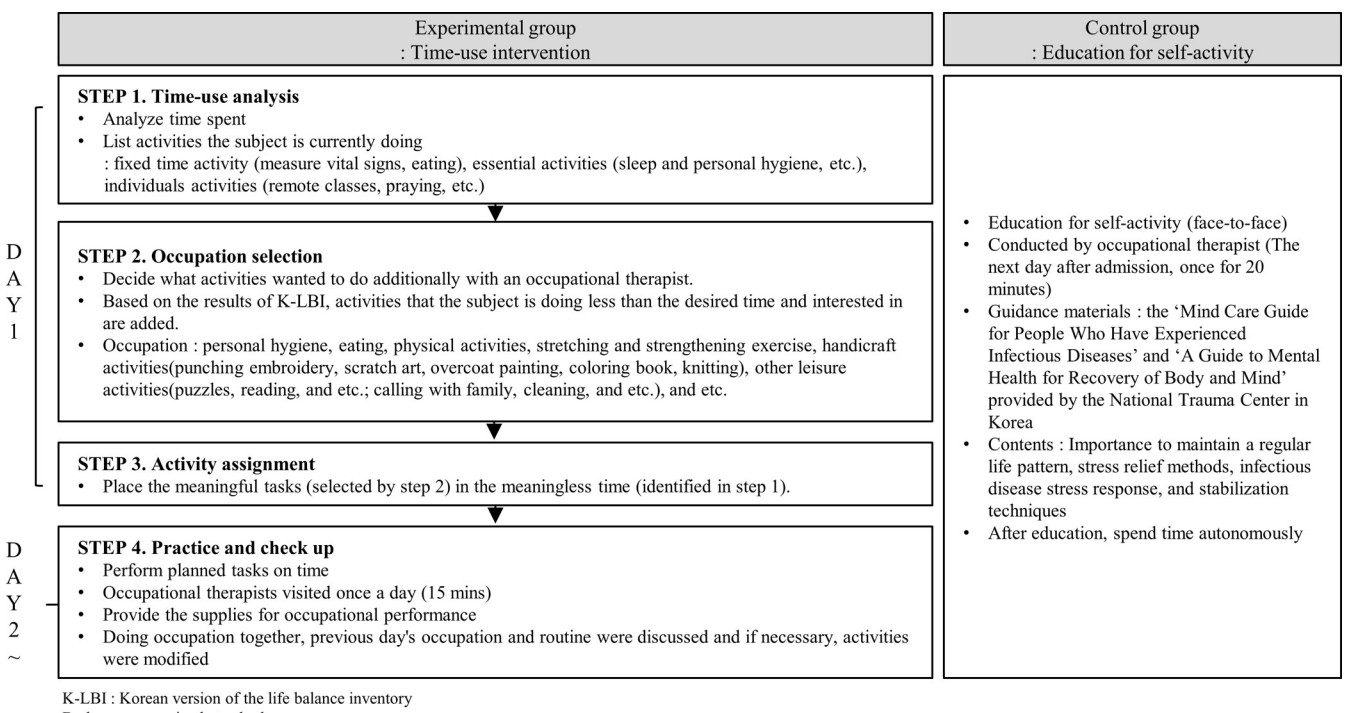

K-LBI : Korean version of the life balance inventory
Both groups received standard care.

**Fig 2. Flow diagram of the time-use intervention.**

1–3 were performed on the first day of the intervention. The next day, the last step was to practice and perform the occupational therapist's intervention. Patients performed occupations on time. The occupational therapist provided materials that were selected in step 2 (punching embroidery, scratch art, etc.) once a day. The materials differed with patient selection. Interviews about the previous day's occupation and routine were also conducted and, if necessary, activities were modified.

### Self-activity education

Self-activity education was conducted once. Educational materials included the "Mind Care Guide for Persons with Infectious Diseases" and "Mental Health Guide for Body and Mind Recovery" provided by the National Trauma Center in Korea, which can be found on the National Trauma Center's website (https://www.nct.go.kr/). Education included the importance of maintaining a regular life pattern, along with stress relief methods and stabilization techniques. The patients were then allowed to spend time autonomously.

### Standard care

Both groups were provided symptomatic treatment for COVID-19-related symptoms and antiviral therapy.

### Outcome measures

**Primary outcome measure.** *Korean Version of Life Balance Inventory (K-LBI)*. The K-LBI was used to measure occupational imbalance in isolated patients with COVID-19. The K-LBI estimates occupational balance by examining the desired usage time and actual usage time for an activity match. It consists of 53 activities (activities of daily living, instrumental activities of daily living, work, play, rest and sleep, leisure, education, and social participation) rated on a 3-point Likert scale ranging from 1 (always less or more than the desired time) to 3 (desired amount of time). The closer the score is to 3, the better the occupational balance. The 53 activities were classified into four subcategories (health, identity, relationship, and challenge) [12].

**Secondary outcome measures.** *Patient Health Questionnaire-9 (PHQ-9)*. The PHQ-9 is a self-reporting tool used to diagnose, screen, monitor, and measure the severity of depression. It includes 9 items, with response options ranging from 0 to 3. Scores range from 0 to 27, with higher scores indicating greater depression severity [13].

*Zung's Self-rating Anxiety Scale (SAS)*. The SAS was used to assess anxiety. It has 20 items rated on a 4-point scale ranging from 1 to 4. It includes 5 items on increasing anxiety levels and 15 items on decreasing anxiety levels [14].

*Korean version of Insomnia Severity Index (ISI-K)*. The ISI-K consists of 7 items regarding the presence and severity of sleep problems. Items are rated on a 5-point Likert scale, and the sum of these scores ranges from 0 to 28 [15].

*Multidimensional State Boredom Scale-8 (MSBS-8)*. The MSBS-8 is a tool for measuring the experience of boredom by shortening the 29-item MSBS to 8 items. Items are rated using a 7-point Likert scale with a total score of 7 to 56, with higher scores reflecting greater boredom [16].

*Fear of COVID-19 scale (FCV-19S)*. The FCV-19S is a self-reported assessment tool used to measure one's fear of COVID-19. The FCV-19S consists of 7 questions, with 4 questions about emotional responses to COVID-19, and 3 questions about physiological responses. It is scored on a 5-point scale ranging from 1 to 5. The total score ranges from 7 to 35, with higher scores indicating a higher fear of COVID-19 [17].

*World Health Organization Quality of Life Assessment Instrument-BRIEF (WHOQOL-BREF)*. The WHOQOL-BREF is a tool used to assess health-related quality of life. It consists of 26 items rated on a 5-point Likert scale ranging from 26 to 130. The items consist of 4 sub-domains (physical health, psychological, social relationships, and environment), overall QOL, and general health. A higher score indicates better QOL [18].

Except for the MSBS-8, outcome measures were measured using the Korean version of the measurement tools. The MSBS-8 was translated into Korean by a research team. All outcome measurements were written in a Google form and assessed by the participants via a tablet PC. Evaluations were conducted after admission and before discharge in both groups.

## Statistical analysis

The sample size was established based on that in previous studies on similar interventions [10,19,20]. A post-hoc power analysis was conducted to examine the statistical power of the group and time differences (Cohen d = 1.21, α error = .05, sample size group 1 = 19, sample size group 2 = 22).

The independent t-test, chi-square test, and two-tailed Fisher exact test were used to compare demographic and clinical characteristics between the two groups. Post-hoc power analysis was used to analyze the K-LBI results using G*Power 3.1.9. Repeated measures analysis of variance was used to measure changes over time between the groups for the pre- and post-repeated measures of the K-LBI, PHQ-9, SAS, ISI-K, MSBS-8, FCV-19S, and WHOQOL-BREF. The unstructured repeated covariance type was used, and fixed effects included group (experimental and control groups) and time (before and after the intervention). The differences between the two groups over time were analyzed by the interaction between the groups using the measurement time points.

Statistical significance was set at $p < .05$. All analyses were performed using SAS statistical software (version 9.4; SAS Institute, Cary, NC, USA).

## Results

### Study population

Fifty patients with COVID-19 were recruited from February 1, 2021, to March 19, 2021 (Fig 1). They were randomly allocated to the experimental group (*n* = 25) or control group (*n* = 25). Nine patients withdrew during the intervention. The oxygen saturation decreased to 90–95% in 8 patients, and 1 patient was discharged early because of 2 consecutive negative real-time polymerase chain reaction test results. The study was completed with 41 participants (19 in the experimental group and 22 in the control group) and an allocation ratio of 82%.

### Patient demographics

The demographic and clinical characteristics of the remaining 41 patients are summarized in Table 1. The experimental group consisted of 7 men and 12 women, with an average age of 47.37±17.61 years and an average isolation period of 13.84±3.15 days. The isolation period varied from 11 to 24 days, depending on the patient's condition. The control group comprised 12 men and 10 women, with an average age of 58.59±13.51 years and an average isolation period of 13.59±3.30 days. The most common clinical symptoms in both groups were fever, cough, sputum, and myalgia. No significant differences were found between the groups in terms of demographic and clinical characteristics (p>.05).

**Table 1. Demographic and clinical characteristics by groups.**

| Classification | | EG (N = 19) | CG (N = 22) | *p*-value |
|---|---|---|---|---|
| Sex, n (%) | Male | 7 (36.84) | 12 (54.55) | .26 |
| | Female | 12 (63.16) | 10 (45.45) | |
| Age (year), M (SD) | | 47.37 (17.61) | 58.59 (13.51) | .24 |
| Duration of isolation (day), M (SD) | | 13.84 (3.15) | 13.59 (3.30) | .81 |
| Education (year), n (%) | Less than middle school | 3 (15.79) | 7 (31.82) | .17 |
| | High school | 4 (21.05) | 7 (31.82) | |
| | More than college | 11 (57.89) | 5 (22.73) | |
| | No response | 1 (5.26) | 3 (13.64) | |
| Work, n (%) | Yes | 16 (84.21) | 13 (59.09) | .08 |
| | No | 3 (15.79) | 9 (40.91) | |
| Clinical symptom, n (%) | Fever (>37.5˚C) | 13 (68.42) | 13 (59.09) | .54 |
| | Myalgia | 11 (57.89) | 13 (59.09) | .94 |
| | Chills | 9 (47.37) | 12 (54.55) | .65 |
| | Cough & sputum | 6 (31.58) | 8 (36.36) | .75 |
| | Fatigue | 7 (36.84) | 6 (27.27) | .51 |
| | Headache | 5 (26.32) | 8 (36.36) | .49 |
| | Sore throat | 8 (42.11) | 7 (31.82) | .50 |
| | Nausea & vomiting | 4 (21.05) | 3 (13.64) | .69 |
| | Diarrhea | 3 (15.79) | 3 (13.64) | 1.00 |
| | Dyspnea | 3 (15.79) | 2 (9.09) | .65 |
| | Hyposmia & hypogeusia | 5 (26.32) | 2 (9.09) | .22 |
| | Dizziness | 3 (15.79) | 2 (9.09) | .65 |
| | Asymptomatic | 2 (10.53) | 1 (4.55) | .59 |

EG = experimental group; CG = control group; M = mean; SD = standard deviation.

The independent t-test, chi-square test, and two-tailed Fisher exact test were performed.

## Occupational balance

In the time × group analysis, the occupational balance items of the experimental group significantly improved compared to those of the control group: K-LBI total score ($F = 14.12$, $p < .001$), health ($F = 10.96$, $p = .002$), challenge and interest ($F = 12.90$, $p < .001$), and identity ($F = 12.29$, $p = .001$; Table 2). The slopes of the K-LBI total score with respect to time in the two groups were opposite (Fig 3). No significant improvement was found in this relationship ($F = 3.77$, $p = .059$).

The occupational balance items of the experimental group improved compared to those before the intervention: K-LBI total score ($p = .0001$), health ($p = .007$), relationship ($p = .017$), challenge and interest ($p = .0001$), and identity ($p < .001$). The occupational balance in the control group decreased compared with that before the intervention. The change in the control group for all occupational balance items was not statistically significant ($p = .05$). The results demonstrated that the power of K-LBI was .96. The post-hoc power analysis showed that our tests met the requirements of power equal to .8.

## Mental health

The results of the PHQ-9, SAS, ISI-K, MSBS-8, and FCV-19S are shown in Table 2. PHQ-9 ($F = 14.23$, $p < .001$), SAS scores ($F = 7.09$, $p = .011$), ISI-K ($F = 31.77$, $p < .0001$), MSBS-8 ($F = 21.63$, $p < .0001$), and FCV-19S ($F = 21.63$, $p = .001$) were significantly improved in the

**Table 2. Comparison of changes in variables within and between the groups.**

| Variables | | Pre | Post | p-value | Cohen d | 95% CI | Time | | Group | | Interaction | | | |
|---|---|---|---|---|---|---|---|---|---|---|---|---|---|---|
| | | M (SD) | M (SD) | | | | F | p-value | F | p-value | F | p-value | Partial omega squared | 95% CI |
| **Occupational balance** | | | | | | | | | | | | | | |
| K-LBI score | EG | 1.88 (0.41) | 2.34 (0.30) | .0001** | -1.28 | -1.39/-1.16 | 5.77 | .021* | 1.77 | .192 | 14.12 | < .001** | 0.11 | 0.02/0.26 |
| | CG | 2.05 (0.40) | 1.95 (0.36) | .378 | 0.27 | 0.15/0.38 | | | | | | | | |
| Health | EG | 2.25 (0.48) | 2.69 (0.41) | .007* | -0.99 | -1.13/-0.84 | 3.20 | .082 | 5.62 | .023* | 10.96 | .002* | 0.10 | 0.02/0.25 |
| | CG | 2.32 (0.38) | 2.19 (0.34) | .209 | 0.36 | 0.26/0.47 | | | | | | | | |
| Relationship | EG | 1.84 (0.71) | 2.15 (0.42) | .017* | -0.53 | -0.71/-0.34 | 1.69 | .200 | 1.12 | .296 | 3.77 | .059 | 0.01 | 0.00/0.12 |
| | CG | 1.83 (0.67) | 1.80 (0.52) | .638 | 0.12 | -0.06/0.29 | | | | | | | | |
| Challenge & interest | EG | 1.76 (0.47) | 2.27 (0.38) | .0001** | -1.20 | -1.34/-1.03 | 5.92 | .020* | 1.54 | .222 | 12.90 | < .001** | 0.10 | 0.02/0.25 |
| | CG | 1.94 (0.48) | 1.84 (0.38) | .452 | 0.23 | 0.10/0.36 | | | | | | | | |
| Identity | EG | 1.86 (0.52) | 2.11 (0.43) | < .001** | -1.12 | -1.27/-0.97 | 5.33 | .026* | 0.54 | .468 | 12.29 | .001* | 0.11 | 0.02/0.26 |
| | CG | 2.11 (0.43) | 2.00 (0.42) | .418 | 0.26 | 0.13/0.38 | | | | | | | | |
| **Mental health** | | | | | | | | | | | | | | |
| PHQ-9 score | EG | 7.11 (6.34) | 4.74 (4.71) | .002* | 0.42 | -1.35/2.20 | 0.00 | .947 | 0.30 | .588 | 14.23 | < .001** | 0.04 | 0.00/0.17 |
| | CG | 3.91 (4.37) | 6.36 (4.58) | .032* | -0.55 | -1.87/0.77 | | | | | | | | |
| SAS score | EG | 39.53 (9.18) | 35.90 (6.22) | .082 | 0.46 | -2.03/2.96 | 0.38 | .544 | 0.98 | .330 | 7.09 | .011* | 0.03 | 0.00/0.16 |
| | CG | 34.73 (6.17) | 37.00 (5.97) | .067 | -0.37 | -2.17/1.42 | | | | | | | | |
| ISI-K score | EG | 11.37 (6.59) | 5.84 (4.44) | < .001** | 0.98 | -0.80/2.77 | 2.08 | .157 | 0.66 | .421 | 31.77 | < .0001** | 0.10 | 0.02/0.25 |
| | CG | 8.41 (6.60) | 11.68 (6.64) | .005 | -0.49 | -2.45/1.46 | | | | | | | | |
| MSBS-8 score | EG | 27.89 (11.94) | 22.53 (7.63) | .030 | 0.54 | -2.65/3.72 | 0.18 | .672 | 0.91 | .345 | 21.63 | < .0001** | 0.09 | 0.01/0.23 |
| | CG | 24.41 (7.49) | 30.86 (8.75) | < .0001** | -0.79 | -3.20/1.61 | | | | | | | | |
| FCV-19S score | EG | 21.37 (7.14) | 17.47 (6.06) | .002* | 0.59 | -1.52/2.69 | 0.32 | .573 | 0.00 | .994 | 12.59 | .001* | 0.06 | 0.00/0.20 |
| | CG | 18.00 (5.56) | 20.82 (5.52) | .031* | -0.51 | -2.15/1.13 | | | | | | | | |
| **Quality of Life** | | | | | | | | | | | | | | |
| WHOQOL-BREF score | EG | 81.47(7.74) | 92.58(11.75) | < .001** | -1.12 | -4.28/2.05 | 8.08 | .007* | 0.54 | .466 | 21.28 | < .0001** | 0.11 | 0.02/0.25 |
| | CG | 86.45(9.52) | 83.82(8.52) | .067 | 0.29 | -2.38/2.96 | | | | | | | | |

EG = experimental group; CG = control group; M = mean; SD = standard deviation; K-LBI = Korean version of the Life Balance Inventory; PHQ-9 = Patient Health Questionnaire-9; SAS = Zung's Self-rating Anxiety Scale; ISI-K = Korean version of the Insomnia Severity Index; MSBS8 = Multidimensional State Boredom Scale-8; FCV-19S = Fear of Coronavirus Disease Scale; WHOQOL-BREF = World Health Organization Quality of Life Assessment Instrument-BREF; Pre = before the intervention; Post = after the intervention.

*p < .05

**p < .001.

experimental group compared with the control group in the time × group analysis. The slopes of the PHQ-9, SAS, ISI-K, MSBS-8, and FCV-19S scores with respect to time were opposite between the two groups (Fig 3). The scores of PHQ (F = 0.00 p = .947), SAS (F = 0.38, p = .544), ISI-K (F = 2.08, p = .157), MSBS-8 (F = 0.18, p = .672), and FCV-19S (F = 0.18, p = .573) did not significantly change over time in either group.

When analyzing the within-group changes, PHQ-9 (p = .002), ISI-K (p < .001), MSBS-8 (p = .030), and FCV-19S (p = .002) were significantly decreased in the experimental group. In contrast, PHQ-9 (p = .032), ISI-K (p = .005), MSBS-8 (p < .0001), and FCV-19S scores (p = .031) were higher in the control group than in the experimental group. In addition, SAS scores decreased in the experimental group and increased in the control group. However, the difference was not statistically significant in either group (p>.05).

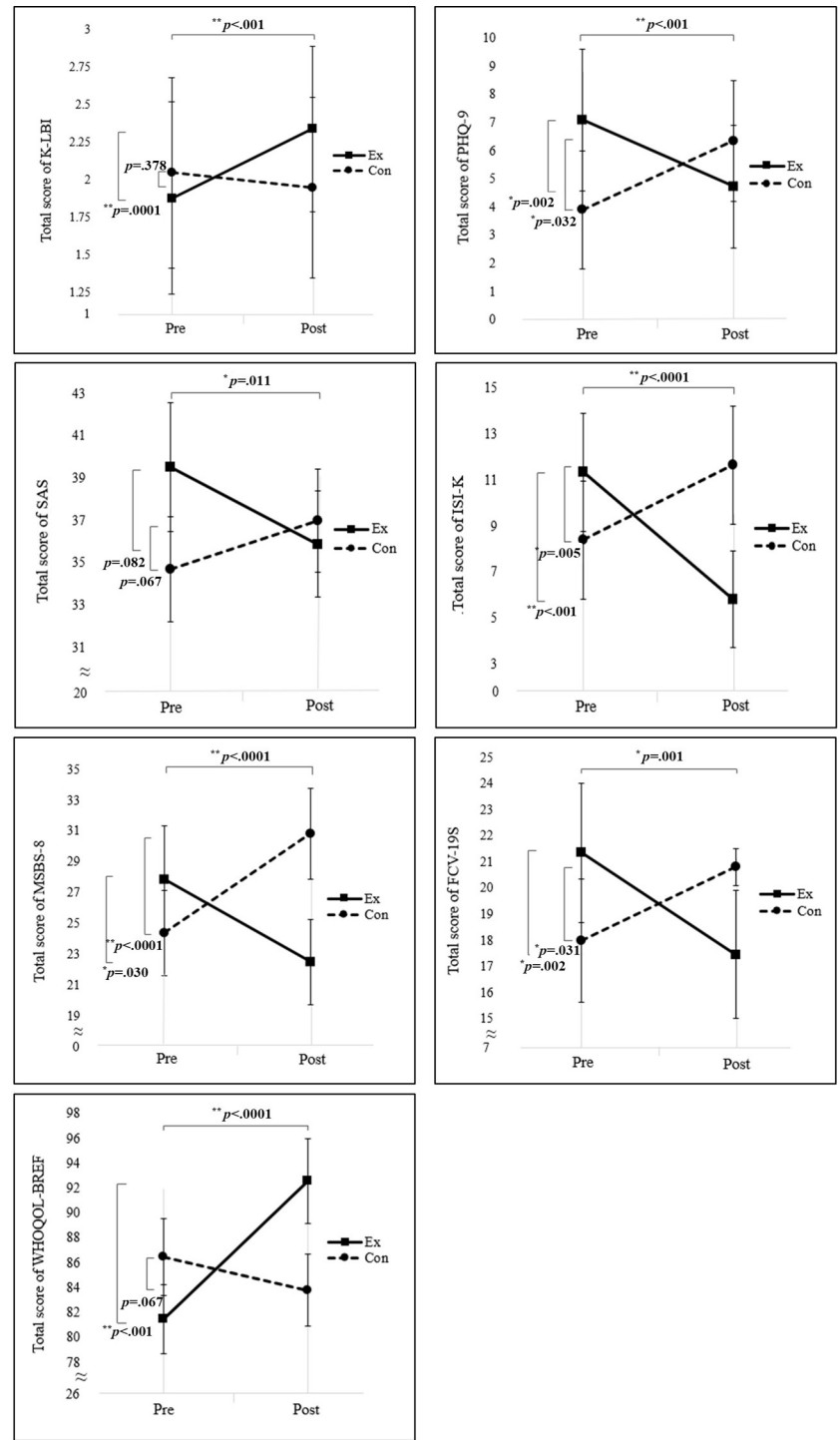

**Fig 3. Scores of the K-LBI, PHQ-9, SAS, ISI-K, MSBS-8, FCV-19S, and WHOQOL-BREF.** The time-use intervention improved occupational balance, mental health (depression, anxiety, insomnia, boredom, fear of COVID-19), and quality of life. K-LBI = Korean version of the Life Balance Inventory; PHQ-9 = Patient Health Questionnaire-9; SAS = Zung's Self-rating Anxiety Scale; ISI-K = Korean version of the Insomnia Severity Index; MSBS8 = Multidimensional State Boredom Scale-8; FCV-19S = Fear of Coronavirus Disease Scale; WHOQOL-BREF = World Health Organization Quality of Life Assessment Instrument-BREF; COVID-19 = coronavirus disease.

## QOL

In the time × group analysis, QOL was significantly improved in the experimental group compared with the control group ($F = 21.28$, $p < .0001$; Table 2). The slopes of the WHOQOL-BREF total score with respect to time in the two groups were opposite (Fig 3). QOL improved in the experimental group after the intervention ($p < .001$), whereas QOL decreased after the intervention in the control group ($p = .067$).

## Discussion

A time-use intervention can improve the occupational balance of patients isolated due to COVID-19. It also improves QOL and reduces psychological problems such as depression, anxiety, insomnia, boredom, and fear. To our knowledge, this study is the first to evaluate the effectiveness of a time-use intervention in patients with COVID-19.

The time-use intervention improved occupational balance in our study. In the time × group analysis, the experimental group showed a significant improvement compared with the control group ($F = 14.12$, $p < .001$). This finding is similar to those of previous studies showing that time-use interventions are effective for individuals with serious mental illness. The time-use intervention group increased their occupational balance by spending more per day on activity than the control group ($p = .05$) [10]. However, in the control group, the mean total K-LBI score decreased from 2.05±0.40 to 1.95±0.3 ($p = .378$). According to previous study results [7], occupational balance was assumed to be disrupted because of COVID-19 confirmation and isolation. In South Korea, patients with confirmed COVID-19 are isolated in the community treatment center [21] or hospital and cannot move outside the room for at least 10 days. Additionally, serious restriction of food intake or activity occurs, which causes disturbances in occupational balance. In a previous study [22], psychological illness due to isolation and the need for screening have been well investigated, but studies on approaches to improve psychological problems have not been conducted. Herein, the time-use intervention was used to help restore occupational balance and improve patients' mental health and QOL.

The time-use intervention helped reduce patients' psychological problems, such as depression, anxiety, insomnia, boredom, and fear. An individual's daily restrictions due to an infectious disease outbreak cause psychological difficulties such as depression, anxiety, and fear [23,24]. Participation in various occupations helps improve mental health [19,25]. Activities during the day can also reduce boredom and improve sleep quality at night. Boredom is also associated with emotional and psychosocial problems [26]. It is a symptom of depression [27], and suicidal thoughts may increase when boredom increases [28]. Moreover, hospitalization can reduce movement in healthy people, and bedridden individuals have limited physical activity [29]. Lack of activity during the day leads to increased daytime sleep and can affect night-time sleep. Encouraging activity during the day can help improve sleep quality [30]. Therefore, it is necessary to fill the day with active activities by encouraging occupational participation during the isolation period.

The time-use intervention helped improve patients' QOL. The total WHOQOL-BREF score of the control group decreased from 86.45±9.52 to 83.82±8.52. The environment of isolation could be a factor that negatively affects patients' QOL [5]. The control group, who received education on self-activity, showed a negative impact on QOL due to isolation over time. However, application of the time-use intervention in the experimental group resulted in improved QOL even though the group was placed in the same isolation environment. QOL is negatively correlated with anxiety and depression [31,32]. In this study, QOL decreased, and anxiety and depression increased in the post-evaluation of the control group.

This study has several limitations. First, it had a small sample size and was conducted at a single center. Therefore, the results cannot be generalized. Further studies should include a larger sample size to validate our findings. However, the results were justified through post hoc analysis and the effect size of each outcome measure. Second, self-reported questionnaires were used as outcome measurements. They rely on participants' honest answers; thus, the possibility of socially desirable responses, recall bias, and misunderstanding of questions cannot be excluded. Third, follow-up after discharge was not performed. Therefore, it was not possible to confirm whether the effect persisted even after discharge. A follow-up evaluation is necessary to confirm this. Fourth, we considered a single-blinded experiment, but this was not feasible because of manpower shortages and the environment of the isolation ward. The non-blinded trial design could have caused performance bias and detection bias. However, we performed the following efforts to reduce the bias. We assigned patients to separate rooms to prevent group mixing, and the same outcome measurements and methods were used in both groups. In addition, two well-trained clinicians collected the data to reduce bias.

Despite these limitations, the time-use intervention was effective in maintaining occupational balance during the isolation period. It also had a positive impact on mental health and QOL. As mental health problems can become serious over time, interventions should begin at the point of isolation. Humans seek satisfaction and lead a happy life by performing occupations [33], and the role of occupational therapists is to provide clients with what they need to achieve a satisfactory QOL [25]. Time-use interventions have been an effective method for providing opportunities for patients with COVID-19 to engage in meaningful occupations during isolation.

## Conclusions

In isolated settings, the time-use intervention improved occupational balance, mental health, and QOL. The isolated environment has had a negative impact on the occupational balance, mental health, and quality of life of patients with COVID-19. In addition, mental health problems can be long-term. Therefore, time-use interventions are needed during the hospitalization period in patients with infectious diseases. This study has clinical significance because it provides an evidence-based intervention that can be applied to isolated patients.

## Supporting information

**S1 Checklist. CONSORT 2010 checklist of information to include when reporting a randomised trial*.**
(DOCX)

**S1 File.**
(PDF)

**S2 File.**
(PDF)

**S3 File.**
(PDF)

## Author Contributions

**Conceptualization:** Jae Hyu Jung, Ji-Hyuk Park.

**Data curation:** Jae Hyu Jung, Jin Young Ko, Ji-Hyuk Park.

**Formal analysis:** Jae Hyu Jung.

**Investigation:** Jae Hyu Jung.

**Methodology:** Jae Hyu Jung, Jin Young Ko, Ickpyo Hong.

**Project administration:** Jin Young Ko.

**Supervision:** Min-Ye Jung, Ji-Hyuk Park.

**Visualization:** Jae Hyu Jung.

**Writing – original draft:** Jae Hyu Jung.

**Writing – review & editing:** Jae Hyu Jung, Ickpyo Hong, Min-Ye Jung, Ji-Hyuk Park.

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
