## [Decision Letter · Decision Letter 0]

4 Sep 2022

PONE-D-22-18523

Effects of Time-use Intervention in Isolated Patients with coronavirus disease 2019: A Randomized Controlled Study

PLOS ONE

Dear Dr. Park,

Thank you for submitting your manuscript to PLOS ONE. After careful consideration, we feel that it has merit but does not fully meet PLOS ONE’s publication criteria as it currently stands. Therefore, we invite you to submit a revised version of the manuscript that addresses the points raised during the review process.

We look forward to receiving your revised manuscript.

Kind regards,

Johannes Stortz

Staff Editor

PLOS ONE

Journal Requirements:

4. Please remove your figures from within your manuscript file, leaving only the individual TIFF/EPS image files, uploaded separately.  These will be automatically included in the reviewers’ PDF.

Additional Editor Comments (if provided):

The manuscript has been evaluated by one reviewer, and their comments are available below.

The reviewer has raised a number of  concerns regarding the methodology, reporting and statistical analysis of this study. 

Could you please revise the manuscript to carefully address the concerns raised?

Reviewers' comments:

Reviewer's Responses to Questions

**Comments to the Author**

1. Is the manuscript technically sound, and do the data support the conclusions?

Reviewer #1: Partly

2. Has the statistical analysis been performed appropriately and rigorously? 

Reviewer #1: No

3. Have the authors made all data underlying the findings in their manuscript fully available?

Reviewer #1: Yes

4. Is the manuscript presented in an intelligible fashion and written in standard English?

Reviewer #1: Yes

5. Review Comments to the Author

Reviewer #1: The study aims to investigate the effect of time-use intervention on occupational balance, mental health, and quality of life in COVID-19 patients who were forced to isolate.

Comments

Abstract

The method section in the abstract requires revision by describing it in a paragraph and not structuring it. Education for self-activity to be mentioned for the control group. Ensure what is written in this section is similar or applied to other section(s) or Figure(s).

Methods

Time use intervention

Line 96, information on allocation concealment and blinding is to be provided. The person who assigned the subjects to intervention and control group to be stated.

Line 106-107, Figure 2 Step 4, for the statement ‘Provide the supplies for occupational performance’, more details are to be provided.

Line 116, standard care to be defined.

Secondary outcome measures

Line 132 -137, for those questionnaires that were not stated in Korean version, it is not clear whether the participants answered the questionnaires via a mix of English and Korean versions questionnaires. The mode of administration whether self-administered or interviewed to be clearly stated.

Statistical analysis

Line 142, For Fisher’s exact test, one or two-tailed test to be stated.

Line 150, for the post-hoc power analysis, more detail is to be provided on how the power is derived when involving two intervention groups.

The reason the sample size calculation was not performed in the study is to be stated.

Results

Line 183-184, 188-189, 196, 205-208, 217 etc some p values were different from Table 2. Also check if p= .001 is the actual value or requires symbol <. Since the p values are available in the table there is no need to use footnote Line 242. Actual p-value to be used in the text even though they are not statistically significant except when summarizing all the p values. This also applies to the text results describing Figure 3 and other sections in the manuscript.

Line 190, the sentence ‘The change in the control group was not statistically significant (p > .05)’ to be revised to ‘The change in the control group for all the occupational balance items was not statistically significant (ps > .05)’

Line 207, 209, the sentence requires revision,

Table 1, the statistical tests to be denoted in table footnote. Having said that, based on CONSORT statement, all statistical tests performed at baseline are to be avoided. Since the symbol % is highlighted after the variable name, all individual symbol % for the figures to be omitted.

Table 2, effect size index and 95% CI could be presented.

Line 242, p<.00 to be revised.

Figure 1, the intervention that the experimental and control group received to be placed in the Figure. The assessment time point/period is to be stated at follow-up. The allocation number for the intervention and control group for those who did not meet the criteria are to be separated and placed in another box. This is to separate the number of patients who received the interventions.

In Figure 2 footnote, to denote both groups received standard care.

Discussion

Line 228, decimal point to be standardised.

Line 268, for the statement small sample size, it requires justification e,g. sample size calculation.

References

List of references did not conform to the journal format.

6. PLOS authors have the option to publish the peer review history of their article (what does this mean?). If published, this will include your full peer review and any attached files.

Reviewer #1: No

---

## [Author Response · Author response to Decision Letter 0]

20 Oct 2022

COMMENTS TO THE AUTHOR:

Reviewer #1: The study aims to investigate the effect of time-use intervention on occupational balance, mental health, and quality of life in COVID-19 patients who were forced to isolate.

- I appreciate for your kind review.

Abstract

1. The method section in the abstract requires revision by describing it in a paragraph and not structuring it. Education for self-activity to be mentioned for the control group. Ensure what is written in this section is similar or applied to other section(s) or Figure(s).

Answer) As your recommendation, method section in the abstract has been rewritten. Also, we have added the contents of self-activity education implemented for the control group. Also, we mentioned time-use intervention and education for self-activity in section of intervention in abstract. 

→ Methods: Randomized controlled single center clinical trial. Forty-one patients (19 in the experimental group, 22 in the control group) with COVID-19 were recruited from February 1 to March 19, 2021. Participants were randomly assigned to receive time-use intervention, or the education for self-activity. Time-use intervention is to plan a daily routine to engage in meaningful occupations. It consists of 4 steps: time use analysis, occupation selection, arrangement of activities, and practice and occupational therapist intervention. The control group is educated for self-activity and spends time autonomously. (Page 2, Line 36-42)

Methods

Time use intervention

2. Line 96, information on allocation concealment and blinding is to be provided. The person who assigned the subjects to intervention and control group to be stated. 

Answer) It was a non-blinded trial because the intervention in this study was not easily blinded. For the allocation concealment, a doctor, not a researcher, performed block randomization. 

→ It is a non-blinded trial, and the participants were assigned in the order of the beds by the physician after he had checked medical state. (Page 5, Line 101-102)

3. Line 106-107, Figure 2 Step 4, for the statement ‘Provide the supplies for occupational performance’, more details are to be provided.

Answer) Ambiguous sentences were corrected. Supplies are provided for the occupation selected in step 2 by occupational therapist. Materials are different with patients’ selection.

→ The participant performed occupations on time. (Page 6, Line 122) The occupational therapist provided materials which were selected in step 2 (punching embroidery, scratch art etc.) once a day. Materials are different with patients’ selection. (Page 6, Line 126-129)

4. 116, standard care to be defined.

Answer) As your comment, we defined standard care. 

→ Both the experimental and the control group were provided symptomatic treatment for COVID-19-related symptoms, and antiviral therapy. (Page 8, Line 141-143)

Secondary outcome measures

5. Line 132 -137, for those questionnaires that were not stated in Korean version, it is not clear whether the participants answered the questionnaires via a mix of English and Korean versions questionnaires. The mode of administration whether self-administered or interviewed to be clearly stated.

Answer) For better understanding, we had cited English version of outcome measures. As your question, we have clarified that the resulting measurement used the Korean version.

→ Except for MSBS-8, outcome measures were used be Korean version. The MSBS-8 was translated into Korean by research team. All outcome measurements were performed by the participants themselves with questionnaires in Google Form. (Page 9, Line 165-167)

Statistical analysis

6. Line 142, For Fisher’s exact test, one or two-tailed test to be stated.

Answer) Two-tailed fisher's exact test was used so we changed the word ‘fisher's exact test’ to ‘two tailed fisher's exact test’. 

→ For the comparison of the demographic and clinical characteristics between the two groups, the independent t-test, the chi-square test, and two-tailed fisher's exact test was used.

 (Page 9, Line 171-175) 

7. Line 150, for the post-hoc power analysis, more detail is to be provided on how the power is derived when involving two intervention groups.

Answer) Added information used for post-hoc power analysis. 

→ Post-hoc power analysis was conducted to examine the statistical power of the group and time differences (Cohen’s d = 1.21, α error = .05, Sample size group 1 = 19, sample size group 2 = 22). (Page 9, Line 183-185)

8. The reason the sample size calculation was not performed in the study is to be stated.

Answer) The sample size was established based on previous studies of similar interventions. To supplement this, post-hoc power analysis was performed.

→ The sample size was established based on previous studies of similar interventions. Post-hoc power analysis was conducted to examine the statistical power of the group and time differences (Cohen’s d = 1.21, α error = .05, Sample size group 1 = 19, sample size group 2 = 22). (Page 9, Line 182-185)

Results

9. Line 183-184, 188-189, 196, 205-208, 217 etc some p values were different from Table 2. Also check if p= .001 is the actual value or requires symbol <. Since the p values are available in the table there is no need to use footnote Line 242. Actual p-value to be used in the text even though they are not statistically significant except when summarizing all the p values. This also applies to the text results describing Figure 3 and other sections in the manuscript.

Answer) The p values written as p<.05 and p<.001 in the manuscript and figure3 have been corrected to the actual values. (Page 12, Line 217. 218. 220. 222. 223. 23. 234. 235. 239. 240. 242-244. 258) Also, Footnote of table 2 in Line 242 (*p<.05, **p<.001) was erased. (Page 27)

10. Line 190, the sentence ‘The change in the control group was not statistically significant (p > .05)’ to be revised to ‘The change in the control group for all the occupational balance items was not statistically significant (ps > .05)’

Answer) As your recommendation, it was revised. 

→ The change in the control group for all the occupational balance items was not statistically significant (p=.05). (Page 12, Line 225-226)

11. Line 207, 209, the sentence requires revision,

Answer) As your recommendation, we revised the sentence.

→ By contrast, PHQ-9 scores (p=.032), ISI-K scores (p=.005), MSBS-8 scores (p<.0001), and FCV-19S scores (p=.031) were increased in the control group. Also, SAS decreased in the experimental group and increased in the control group. However, the difference was not statistically significant in both groups (p>.05). (Page 13, Line 249-251)

12. Table 1, the statistical tests to be denoted in table footnote. Having said that, based on CONSORT statement, all statistical tests performed at baseline are to be avoided. Since the symbol % is highlighted after the variable name, all individual symbol % for the figures to be omitted.

Answer) The independent t test, chi-square, and Fisher's extract test were performed for descriptive data analysis. As your comment, we added on footnote of table 1 about statistical test. According to the CONSORT statement, table 1 has been modified with values that have not been tested for significance (p value of duration of isolation: .62 to .84). Also, % was deleted. (Page 25)

13. Table 2, effect size index and 95% CI could be presented.

Answer) We added effect size index and 95% CI on Table 2. We calculated cohen`s d within group and partial omega squared in interaction. (Page 26-27)

14. Line 242, p<.00 to be revised.

Answer) As your first comment in the result, footnote of table 2 in Line 242 (*p<.05, **p<.001) was erased. (Page 27)

15. Figure 1, the intervention that the experimental and control group received to be placed in the Figure. The assessment time point/period is to be stated at follow-up. The allocation number for the intervention and control group for those who did not meet the criteria are to be separated and placed in another box. This is to separate the number of patients who received the interventions.

Answer) The assessment period was stated at follow-up in Figure 1. However, it is difficult to write an accurate time because the hospitalization period is different for each patient depending on the medical condition. We changed "post assessment" to "post assessment at discharge". Also, descriptions of patients who discontinued during the experiment were separated and placed in another box. 

16. In Figure 2 footnote, to denote both groups received standard care.

Answer) Added footnote in Figure 2; Both groups received standard care.

iscussion

17. Line 228, decimal point to be standardised.

Answer) It has been standardized to describe up to 3 decimal places. W replaced 0.37 with .378. (Page 11, Line 242)

→ However, in the control group, the mean of the total K-LBI score decreased from 2.05±0.40 to 1.95±0.3 (p=.378). (Page 14, Line 268-269)

18. 268, for the statement small sample size, it requires justification e,g. sample size calculation.

Answer) We wrote that because the absolute number of samples was small (experimental group, n =19). However, it was justified through post-hoc analysis (Page 12, Line 227-228) and effect size of each outcome measures (Page 26-27). 

→ However, it was justified through post-hoc analysis and effect size of each outcome measures. (Page 19, Line 311-312)

References

19. List of references did not conform to the journal format.

Answer) We modified journal format suitable for Plos one's guidelines.

---

## [Decision Letter · Decision Letter 1]

21 Dec 2022

PONE-D-22-18523R1Effects of Time-use Intervention in Isolated Patients with coronavirus disease 2019: A Randomized Controlled StudyPLOS ONE

Dear Dr. Park,

Thank you for submitting your manuscript to PLOS ONE. After careful consideration, we feel that it has merit but does not fully meet PLOS ONE’s publication criteria as it currently stands. Therefore, we invite you to submit a revised version of the manuscript that addresses the points raised during the review process.

We look forward to receiving your revised manuscript.

Kind regards,

Jianhong Zhou

Staff Editor

PLOS ONE

Journal Requirements:

Reviewers' comments:

Reviewer's Responses to Questions

**Comments to the Author**

1. If the authors have adequately addressed your comments raised in a previous round of review and you feel that this manuscript is now acceptable for publication, you may indicate that here to bypass the “Comments to the Author” section, enter your conflict of interest statement in the “Confidential to Editor” section, and submit your "Accept" recommendation.

Reviewer #1: All comments have been addressed

Reviewer #2: All comments have been addressed

Reviewer #3: (No Response)

2. Is the manuscript technically sound, and do the data support the conclusions?

Reviewer #1: Partly

Reviewer #2: Yes

Reviewer #3: Yes

3. Has the statistical analysis been performed appropriately and rigorously? 

Reviewer #1: Yes

Reviewer #2: I Don't Know

Reviewer #3: Yes

4. Have the authors made all data underlying the findings in their manuscript fully available?

Reviewer #1: Yes

Reviewer #2: Yes

Reviewer #3: No

5. Is the manuscript presented in an intelligible fashion and written in standard English?

Reviewer #1: Yes

Reviewer #2: No

Reviewer #3: Yes

6. Review Comments to the Author

Reviewer #1: Table 1 footnote, typo error ' Fisher's extract test'.

Reviewer #2: One more thorough review of the English language editing is required.

Please clarify the primary objective and secondary objective in the objective part of the abstract.

Please slightly describe the occupation imbalance/balance and time use intervention at its first use in the introduction. It is not clear while reading the introduction.

How was the informed consent of the patient taken if the written form were waived? Write about the approval of the hospital to undertake the study.

Describe a little bit about the study setting. What is the duration of the patient staying at the hospital? On what day of residing was the intervention taken, and how long and how many times was the intervention given is not clear in detail? It is unclear about the duration between pre-and post-assessment. Provide detailed information on the time length taken in each step from the start of enrolment to final study completion.

Describe how the medical checkup criteria of the patient were checked and confirmed before undertaking the study.

Describe all the tools used in the study with the rationality of choosing them in measuring the outcome in the method part shortly.

Sample size calculation and the inclusion criteria would be good to cover in the same subheading. Please cite the paper referenced to calculate the sample for this study.

Also, add the limitation of a non-blinded trial in the limitation part.

The method and procedure of data collection and the questionnaire format need to be described in little detail in the method section.

Reviewer #3: The authors evaluated a new protocol for time-use among patients suffering from infectious diesases requiring isolation which is very important in this current situation when the world is fighting for prevention and control of the ongoing COVID-19 pandemic. If enough evidence is generated through the intervention trials for improving mental health, occupational balance and quality of life during such crisis where patients require to stay isolated for long period of time from family and friend, it can be impactful for improving patient’s compliance to treatment even during convalescent period.

The manuscript may be considered for publication after addressing the following comments.

Methods:

Line 79: What assumptions were made for including 50 patients for this trial? Justification for this sample size need to be addressed. To study effectiveness of a new intervention protocol, a considerable number of samples needs to be included to provide a significant result of the analysis.

Line 80: it is not clear if the patients were isolated in single cabin or ward. How were the study participants followed-up?

Line 83: What measures taken to avoid bias? If a patient assigned to control group gets into contact with intervention group, there is a risk that control group participants may adapt some of the tasks assigned to intervention group. Hence was any measure taken to avoid the introduction of this bias?

Line 97: How were the participants instructed or briefed about the intervention protocol. Was there any formal training provided to the participants in the intervention arm on the procedures to comply the intervention protocol?

Line 101: How is the occupation selection done? Please define the steps to clarify.

Results

Line 207: Delete “was” as it is abundant in this sentence.

Table 2- The statistical tests used in this analysis should be mentioned in the footnote with the appropriate level of significance.

Discussion:

Line 229: Authors can emphasize on the scoring range found from previous studies that correlates with the level of stress.

Line 261-266: Elaborate the discussion on how the QOL scores differ in different settings of isolation. For example, isolating in a single cabin, ward, or institute with having or not having activities involved during the isolation period.

Conclusion:

Line 291-294: Authors should emphasize the prospective impact of this new developed protocol of the time-use intervention as well as way forward in the field of mental health and challenges.

7. PLOS authors have the option to publish the peer review history of their article (what does this mean?). If published, this will include your full peer review and any attached files.

Reviewer #1: No

Reviewer #2: No

Reviewer #3: No

---

## [Author Response · Author response to Decision Letter 1]

4 Feb 2023

Review Comments to the Author

Reviewer #1: 

1. Table 1 footnote, typo error ' Fisher's extract test'.

Answer> As your recommendation, we modified 'Fisher's extract test' to 'fisher's exact test. Thanks for finding the typo error. 

- Independent t test, Chi-square, and two-tailed fisher's exact test' were performed (Page 22, Line 519)

Reviewer #2: 

1. One more thorough review of the English language editing is required.

Answer> As your recommendation, I reviewed the English language edit again.

2. Please clarify the primary objective and secondary objective in the objective part of the abstract. 

Answer> As your recommendation, we clarified the primary objective and secondary objective in the objective part of the abstract. The primary objective is to find out the impact of time use intervention on the occupational balance of COVID-19 patients, and the secondary objective is the impact on mental health and quality of life.

- The primary objective of this study was to determine the effectiveness of time-use intervention on occupational balance in isolated COVID-19 patients. The impact on secondary outcomes including mental health and quality of life were also assessed. (Page 2, Line 26-29)

3. Please slightly describe the occupation imbalance/balance and time use intervention at its first use in the introduction. It is not clear while reading the introduction. 

Answer> As your recommendation, we added a brief explanation of occupation imbalance/balance and time-use intervention in the introduction and rearranged the introduction to make it easier to understand.

- Occupation balance is defined as the organization ….. occupation area to maintain occupation balance [8, 9] (Page 4, Line 71-82)

4. How was the informed consent of the patient taken if the written form were waived? Write about the approval of the hospital to undertake the study.

Answer> As your comment, we added an explanation for obtaining informed consent from the patient in method section. I have attached the Google form that I actually used to obtain consent below to help your understanding. The green section is research information and contents. If patients had voluntary participated in the study, they pressed the "I agree" button in red section. Also, we conducted the study in compliance with the hospital's ethical regulations. 

- As an alternative to written consent, it was designed to proceed with the pre-evaluation when online consent was given by clicking the “I agree” button on the Google form with tablet PC.. (Page 6, Line 124-126)

5. Describe a little bit about the study setting. What is the duration of the patient staying at the hospital? On what day of residing was the intervention taken, and how long and how many times was the intervention given is not clear in detail? It is unclear about the duration between pre-and post-assessment. Provide detailed information on the time length taken in each step from the start of enrolment to final study completion. 

Answer> The isolation period (hospitalization period) of the patients varied from 11 to 24 days depending on the patient's condition. Inconsistent length of hospital stay may affect the results, but there was no significant difference in the duration of isolation between the two groups as shown in Table 1. Time-use intervention was performed over 7 days depending on hospitalization period in the experimental group. Education for self-activity is provided once and allows the patient to act autonomously during hospitalization. Both interventions are premised on the patient's activity during hospitalization. Also, evaluations were conducted once each after admission and before discharge for both groups. Therefore, duration between pre-and post-assessment is different for each patient. Duration between pre-and post-assessment is similar with the hospitalization period. Finally, Steps 1 to 3 of time-use intervention were performed on the first day of intervention. As your comment, we added that contents to the manuscript. 

- The experimental group received time-use intervention and the control group received education for self-activity. Interventions were applied every day during the isolation period. All participants received the intervention individually. (Page 7, Line 146-148)

- All outcome measurements were written in a google form and performed by the participants themselves with google form of table PC. Evaluations were conducted after admission and before discharge for both groups. (Page 11, Line 237-239)

- The isolation period of the patients varied from 11 to 24 days depending on the patient's condition. (Page 13, Line 271-272)

- Education for self-activity was conducted once in the control group. (Page 8, Line 169)

- Time-use intervention was performed over 7 days depending on length of hospitalization in the experimental group. (Page 7, Line 150)

- Patients and therapist place the meaningful tasks in the meaningless time which selected in step 2. Also, patients and therapist create a timetable. Steps 1 to 3 were performed on the first day of intervention. The next day, the last step is to practice and occupational therapist intervention. (Page 8, Line 157-162)

6. Describe how the medical checkup criteria of the patient were checked and confirmed before undertaking the study.

Answer> As your recommendation, we described the medical checkup criteria of the patient were checked before the study. 

- The ward nurse routinely conducts medical checkups such as chest x-ray, blood pressure, blood test, oxygen saturation, and BT for all inpatients regardless of research. This is to determine the severity of pneumonia and whether it is hemodynamically stable. Only patients who agreed to the study had their medical state reviewed by the medical staff of this study. 

(Page 5-6, Line 106-110)

7. Describe all the tools used in the study with the rationality of choosing them in measuring the outcome in the method part shortly.

Answer> As your recommendation, we added explanation of secondary outcome measures shortly.

- Patient Health Questionnaire-9 (PHQ-9) …… A higher score indicates a better QOL [22].

(Page 9-10, Line 195-234)

8. Sample size calculation and the inclusion criteria would be good to cover in the same subheading. Please cite the paper referenced to calculate the sample for this study.

Answer> As your recommendation, we transferred sentence about sample size calculation from section for Statistical analysis to same section with inclusion criteria. Also, we cited the reference previous study about sample size. 

- The sample size was….. LBI using G*Power 3.1.9. (Page 6, Line 127-132)

9. Also, add the limitation of a non-blinded trial in the limitation part.

Answer> As your recommendation, we added limitation about lacking blinding. Non-blinded trial could cause performance and detection bias. We assigned groups separately and use same outcome measurement and methods to reduce bias. In addition, data collection was conducted by two well-trained clinicians with more than 5 years of experience. 

- Fourth, we considered a single blinded experiment, but it was not feasible due to manpower shortages and the environment of the isolation ward. This study is non-blinded trial could cause performance bias and detection bias. However, we did the following efforts to reduce the bias. We assigned patients to separate rooms to prevent group mixing, and same outcome measurement and methods were used for both groups. Also, two well-trained clinicians conducted data collection to reduce bias. (Page 17, Line 365-370)

10. The method and procedure of data collection and the questionnaire format need to be described in little detail in the method section.

Answer> As your comment, we added that contents to the method. Evaluations were conducted after admission and before discharge for both groups. All outcome measures were written in a google form, and patients evaluated with google form of table PC. This is a method often used in the non-face-to-face era. We attach some of the google forms that we used to help understand. Yellow section is outcome measurements of google form. 

- Evaluations were conducted after admission and before discharge for both groups. All outcome measures were written in a google form, and patients evaluated with google form of table PC. (Page 11, Line 237-239)

Reviewer #3: 

The authors evaluated a new protocol for time-use among patients suffering from infectious diesases requiring isolation which is very important in this current situation when the world is fighting for prevention and control of the ongoing COVID-19 pandemic. If enough evidence is generated through the intervention trials for improving mental health, occupational balance and quality of life during such crisis where patients require to stay isolated for long period of time from family and friend, it can be impactful for improving patient’s compliance to treatment even during convalescent period.

Answer> I appreciate for your kind review.

The manuscript may be considered for publication after addressing the following comments.

Methods:

Line 79: What assumptions were made for including 50 patients for this trial? Justification for this sample size need to be addressed. To study effectiveness of a new intervention protocol, a considerable number of samples needs to be included to provide a significant result of the analysis.

Answer> Previous studies were referred to for setting the sample size. Citations of previous studies referenced have been added. In addition, the sample size was justified through post analysis.

- The sample size was….. LBI using G*Power 3.1.9. (Page 6, Line 127-132)

Line 80: it is not clear if the patients were isolated in single cabin or ward. How were the study participants followed-up? 

Answer> As your comments, we added information about ward types. All patients were isolated in ward for 4 patients. The same researcher followed up during hospitalization. We couldn't follow up after discharge.

- The two groups were assigned to wards separately so that they did not mix. (Page 5, Line 103)

Line 83: What measures taken to avoid bias? If a patient assigned to control group gets into contact with intervention group, there is a risk that control group participants may adapt some of the tasks assigned to intervention group. Hence was any measure taken to avoid the introduction of this bias?

Answer> This study was randomization control trial, but we could not be performed double blinded. So, it may cause bias. However, we tried to avoid bias through the following measures. Wards consisted of a hospital room used by patients who were not registered in the study and the experimental group, patients who were not registered in the study and the control group. The hospital that conducted the study operated 20 four-person rooms. As your comments, we added sentence as follow.

- The two groups were assigned to wards separately so that they did not mix. (Page 5, Line 103)

Line 97: How were the participants instructed or briefed about the intervention protocol. Was there any formal training provided to the participants in the intervention arm on the procedures to comply the intervention protocol? 

Answer> 

Time use intervention : The time-use intervention flow is explained through data similar with Figure 2 before starting. Occupational therapist and patients made timetable together. We attached intervention flow and timetable that we actually used for better understanding. Fill in the gray areas with the occupations selected in Step 2. Participants look at the timetable and implement the planned occupations. The therapist confirmed daily occupation performance.

Education for self-activity : We used the ‘Mind Care Guide for Persons with Infectious Diseases’ and the ‘Mental Health Guide for Body and Mind Recovery’ provided by the National Trauma Center in Korea as educational materials. Education includes the importance to maintain a regular life pattern along with stress relief methods and stabilization techniques. The picture is part of the educational material. It can be found on the National Trauma Center website (https://www.nct.go.kr/).

- Educational materials were used the ‘Mind Care Guide for Persons with Infectious Diseases’ and the ‘Mental Health Guide for Body and Mind Recovery’ provided by the National Trauma Center in Korea. It can be found on the National Trauma Center website (https://www.nct.go.kr/). Education includes the importance to maintain a regular life pattern along with stress relief methods and stabilization techniques. (Page 8, Line 169-172)

Line 101: How is the occupation selection done? Please define the steps to clarify.

Answer> Occupation selection is based on the results of K-LBI which our primary outcome measurement. There are 53 occupations in K-LBI. Activities that the subject is doing less than the desired time and activities that the subject is interested in are added. We attach a part of the English version of LBI for better understanding.

- Occupation selection is based on the results of K-LBI. There are 53 occupations in K-LBI. Activities that the subject is doing less than the desired time and activities that the subject is interested in are added. (Page 7, Line 153-156)

Results

Line 207: Delete “was” as it is abundant in this sentence.

Answer> As your recommendation, we deleted “was” in line 207. 

- The change in the control group for all the occupational balance items not statistically significant (p=.05). (Page 13, Line 289)

Table 2- The statistical tests used in this analysis should be mentioned in the footnote with the appropriate level of significance.

Answer> As your recommendation, we added mention in the table 2, footnote, and figure3 with the appropriate level of significance. We set the level of significance to *<0.05, **<0.001. (Page 23-24, Figure3)

Discussion:

Line 229: Authors can emphasize on the scoring range found from previous studies that correlates with the level of stress. 

Answer> ‘Correlates’ has been modified to ‘similar’. I am sorry to there was an error as English is not the main language of us. And I added the contents of the previous study.

- This finding similar with those of previous studies showing that time-use intervention is effective for persons with serious mental illness. Time-use intervention group increased their occupational balance by spending more per day in activity than the control group (p = .05) [11]. (Page 15, Line 323-326)

Line 261-266: Elaborate the discussion on how the QOL scores differ in different settings of isolation. For example, isolating in a single cabin, ward, or institute with having or not having activities involved during the isolation period.

Answer> The setting of the patients was the same as ward. We think your comment is very interesting. Unfortunately, we could not identify each patient's having or not having activities due to the lack of manpower and isolation environment. However, since the K-LBI score and QOL score of the experimental group are higher than the control group, it can be indirectly interpreted that having activities as much as desired is helpful for QOL.

Conclusion:

Line 291-294: Authors should emphasize the prospective impact of this new developed protocol of the time-use intervention as well as way forward in the field of mental health and challenges.

Answer> As your recommendation, we revised conclusion section. 

- Isolated environment has had a negative impact on the occupational balance, mental health, and quality of life of COVID-19 patients. In addition, mental health problems can be long-term.

- Therefore, time-use intervention is needed to provide from the hospitalization period in patients with infectious diseases. (Page 18, 387-388)

---

## [Decision Letter · Decision Letter 2]

2 Mar 2023

PONE-D-22-18523R2Effects of Time-use Intervention in Isolated Patients with coronavirus disease 2019: A Randomized Controlled StudyPLOS ONE

Dear Dr. Park,

Thank you for submitting your manuscript to PLOS ONE. After careful consideration, we feel that it has merit but does not fully meet PLOS ONE’s publication criteria as it currently stands. Therefore, we invite you to submit a revised version of the manuscript that addresses the points raised during the review process.

The reviewer feels that you addressed all previous comments. However, we have noted that some journal requirements have not been fulfilled which need to be addressed in order to proceed with your manuscript. Please find these requests listed below under 'Additional Editor Comments'.

We look forward to receiving your revised manuscript.

Kind regards,

Alex Schaefer, PhD

Associate Editor

PLOS ONE

Journal Requirements:

Additional Editor Comments:

<ol1 style="list-style-type:decimal;"> <li>  

Please note that you must upload a completed CONSORT checklist as a supporting information file. Blank copies of these documents and information regarding CONSORT can be found via the following link: http://www.consort-statement.org/.

 <li>  

Please upload a copy of the original trial study protocol as a supporting information file. By the study protocol, we mean the complete and detailed plan for the conduct and analysis of the trial that the ethics committee approved before the trial began. Please send this in the original language. If this is in a language other than English, please also provide a translation. Please detail any deviations from this study protocol in the Methods section of your manuscript. Your study protocol will be made available to the editors and reviewers, and will be published as supporting information with your manuscript if accepted for publication. (If you do not agree to this, we will not be able to publish your manuscript). If you have formally published a study protocol for your trial in a journal then you should cite this in your manuscript, but you still need to send us the original document.

 <li>  

We suggest you thoroughly copyedit your manuscript for language usage, spelling, and grammar. If you do not know anyone who can help you do this, you may wish to consider employing a professional scientific editing service. Whilst you may use any professional scientific editing service of your choice, PLOS has partnered with both American Journal Experts (AJE) and Editage to provide discounted services to PLOS authors. Both organizations have experience helping authors meet PLOS guidelines and can provide language editing, translation, manuscript formatting, and figure formatting to ensure your manuscript meets our submission guidelines. To take advantage of our partnership with AJE, visit the AJE website (http://learn.aje.com/plos/) for a 15% discount off AJE services. To take advantage of our partnership with Editage, visit the Editage website (www.editage.com) and enter referral code PLOSEDIT for a 15% discount off Editage services. If the PLOS editorial team finds any language issues in text that either AJE or Editage has edited, the service provider will re-edit the text for free. Upon resubmission, please provide the following:

 <ol1 style="list-style-type:lower-alpha;"> <li>  

 <li>  

 <li>  

Reviewers' comments:

Reviewer's Responses to Questions

**Comments to the Author**

1. If the authors have adequately addressed your comments raised in a previous round of review and you feel that this manuscript is now acceptable for publication, you may indicate that here to bypass the “Comments to the Author” section, enter your conflict of interest statement in the “Confidential to Editor” section, and submit your "Accept" recommendation.

Reviewer #1: All comments have been addressed

2. Is the manuscript technically sound, and do the data support the conclusions?

Reviewer #1: (No Response)

3. Has the statistical analysis been performed appropriately and rigorously? 

Reviewer #1: (No Response)

4. Have the authors made all data underlying the findings in their manuscript fully available?

Reviewer #1: (No Response)

5. Is the manuscript presented in an intelligible fashion and written in standard English?

Reviewer #1: (No Response)

6. Review Comments to the Author

Reviewer #1: (No Response)

7. PLOS authors have the option to publish the peer review history of their article (what does this mean?). If published, this will include your full peer review and any attached files.

Reviewer #1: No

---

## [Author Response · Author response to Decision Letter 2]

12 Mar 2023

Journal Requirements:

1. Please review your reference list to ensure that it is complete and correct. 

Answer) Reference has been corrected. Two references (previous numbers: 11 and 28) were deleted as duplicates, and two references (previous numbers: 12 and 15) were deleted as retracted articles. Also, reference 19 (existing number: 21) written in Korean has been revised into English.

Additional Editor Comments:

1. Please note that you must upload a completed CONSORT checklist as a supporting information file. Blank copies of these documents and information regarding CONSORT can be found via the following link: http://www.consort-statement.org/.

Answer) We uploaded the CONSORT checklist to the supporting information file. 

2. Please upload a copy of the original trial study protocol as a supporting information file. 

Answer) We uploaded the original trial study protocol to the supporting information file. 

3. We suggest you thoroughly copyedit your manuscript for language usage, spelling, and grammar. 

Answer) We reviewed and revised language usage, spelling, and grammar once again.

---

## [Editor Report · Decision Letter 3]

15 Mar 2023

PONE-D-22-18523R3

Effects of a Time-use Intervention in Isolated Patients with Coronavirus Disease 2019: A Randomized Controlled Study

PLOS ONE

Dear Dr. Park,

Thank you for submitting your manuscript to PLOS ONE. After careful consideration, we feel that it has merit but does not fully meet PLOS ONE’s publication criteria as it currently stands. Therefore, we invite you to submit a revised version of the manuscript that addresses the points raised during the review process.

We have noted that the trial study protocol you have provided is not the original version as submitted to the ethics committee. The document you have provided is taken from the clinical trial registry website. 

In order to proceed with your manuscript, please upload a copy of your original full length trial study protocol as a supporting information file. 

By the study protocol, we mean the complete and detailed plan for the conduct and analysis of the trial that the ethics committee approved before the trial began. Please note that this document should typically contain a background section and a detailed description of the methods used including information about dates of recruitment, participant characteristics, inclusion/exclusion criteria, detailed descriptions of the protocols/interventions utilized, plans for outcome assessment and data collection/analysis, and references. 

Please send the study protocol in the original language. If this is in a language other than English, please also provide a translation. Please detail any deviations from this study protocol in the Methods section of your manuscript. Your study protocol will be made available to the editors and reviewers, and will be published as supporting information with your manuscript if accepted for publication. (If you do not agree to this, we will not be able to publish your manuscript). If you have formally published a study protocol for your trial in a journal then you should cite this in your manuscript, but you still need to send us the original document.

We look forward to receiving your revised manuscript.

Kind regards,

Alex Schaefer, PhD

Associate Editor

PLOS ONE
---

## [Author Response · Author response to Decision Letter 3]

18 Apr 2023

1. Our study protocol was written in Korean. Therefore, the English translation is attached together.

2. Reference has been corrected. Two references (previous numbers: 11 and 28) were deleted as duplicates, and two references (previous numbers: 12 and 15) were deleted as retracted articles. Also, reference 19 (existing number: 21) written in Korean has been revised into English.

---

## [Editor Report · Decision Letter 4]

31 May 2023

Effects of a Time-use Intervention in Isolated Patients with Coronavirus Disease 2019: A Randomized Controlled Study

PONE-D-22-18523R4

Dear Dr. Park,

We’re pleased to inform you that your manuscript has been judged scientifically suitable for publication and will be formally accepted for publication once it meets all outstanding technical requirements.

Kind regards,

Dario Ummarino, PhD

Staff Editor

PLOS ONE
---

## [Editor Report · Acceptance letter]

15 Jun 2023

PONE-D-22-18523R4 

Effects of a Time-use Intervention in Isolated Patients with Coronavirus Disease 2019: A Randomized Controlled Study 

Dear Dr. Park:

I'm pleased to inform you that your manuscript has been deemed suitable for publication in PLOS ONE. Congratulations! Your manuscript is now with our production department. 

Kind regards, 

on behalf of

Dr Dario Ummarino, PhD 

Staff Editor

PLOS ONE